# Integrating Genome-Wide Association Study with Transcriptomic Analysis to Predict Candidate Genes Controlling Storage Root Flesh Color in Sweet Potato

Yi Liu [1,†], Rui Pan [2,†], Wenying Zhang [2], Jian Lei [1], Lianjun Wang [1], Shasha Chai [1], Xiaojie Jin [1], Chunhai Jiao [1,*] and Xinsun Yang [1,*]

[1] Food Crops Institute, Hubei Academy of Agricultural Sciences, Wuhan 430064, China; 201973042@yangtzeu.edu (Y.L.); leijian2006@hbaas.com (J.L.); wanglianjun@hbaas.com (L.W.); chaishasha2008@hbaas.com (S.C.); xiaojiejin@hbaas.com (X.J.)
[2] Research Center of Crop Stresses Resistance Technologies/Hubei Collaborative Innovation Center for Grain Industry, Yangtze University, Jingzhou 434025, China; 201973044@yangtzeu.edu (R.P.); wyzhang@yangtzeu.edu.cn (W.Z.)
\* Correspondence: jiaoch@hotmail.com (C.J.); yangshao2021@hbaas.com (X.Y.)
† These authors contributed equally to this work.

**Abstract:** Sweet potato is a hexaploid heterozygote with a complex genetic background, self-pollination infertility, and cross incompatibility, which makes genetic linkage analysis quite difficult. Genome-wide association studies (GWAS) provide a new strategy for gene mapping and cloning in sweet potato. Storage root flesh color (SRFC) is an important sensory evaluation, which correlates with storage root flesh composition, such as starch, anthocyanin, and carotenoid. We performed GWAS using SRFC data of 300 accessions and 567,828 single nucleotide polymorphism (SNP) markers. Furthermore, we analyzed transcriptome data of different SRFC varieties, and conducted real-time quantitative PCR (qRT-PCR) to measure the expression level of the candidate gene in purple and non-purple fleshed sweet potato genotypes. The results showed that five unique SNPs were significantly ($-\log_{10}P > 7$) associated with SRFC. Based on these trait-associated SNPs, four candidate genes, g55964 (*IbF3'H*), g17506 (*IbBAG2-like*), g25206 (*IbUGT-73D1-like*), and g58377 (*IbVQ25-isoform X2*) were identified. Expression profiles derived from transcriptome data and qRT-PCR analyses showed that the expression of g55964 in purple-fleshed sweet potato was significantly ($p < 0.01$) higher than that of non-purple fleshed sweet potato. By combining the GWAS, transcriptomic analysis and qRT-PCR, we inferred that *g55964* is the key gene related to purple formation of storage root in sweet potato. Our results lay the foundation for accelerating sweet potato genetic improvement of anthocyanin through marker-assisted selection.

**Keywords:** sweet potato; genome-wide association studies; storage root flesh color; transcriptomic analysis

## 1. Introduction

Sweet potato (*Ipomoea batatas* (L.) Lam) is a hexaploid (2n = 6× = 90) species, rich in high calories, proteins, vitamins, and minerals. It is the seventh most important crop in the world and the fourth most significant crop cultivated in China [1]. Sweet potato breeding programs worldwide are integrating sensory characteristics to provide global markets with improved cultivars [2–4]. SRFC is identified as a key driver of consumers' sensory and hedonic expectations before tasting [5].

SRFC fluctuates in thousands of ways, white, light yellow, yellow, orange, light red, pink, red, fuchsia, purple, deep purple, and dark purple viewed by human eyes. According to SRFC, sweet potato can be divided into white, yellow or orange, and purple cultivars. According to chemical composition, sweet potato can be classified into starch, anthocyanin,

and carotenoid cultivars. The major pigments in sweet potato are anthocyanins and carotenoids, which are responsible for sweet potato's storage root colors [6]. There is a correlation between color and substance content of storage root flesh: the purple-fleshed has the highest anthocyanins content, and orange-fleshed has the highest carotene, while white-fleshed have the lowest [7,8]. Anthocyanin and carotene contents varied widely among the sweet potato varieties with different flesh color.

Anthocyanins are responsible for the shiny orange, pink, red, purple, and blue colors in the flowers and fruits of some plants [9]. Anthocyanins are initially synthesized in the cytoplasm, and transported to vacuoles, where they are sequestered to form colors in different plant organs [10]. Vesicles, glutathione S-transferase, and membrane transporters are involved in the transport of anthocyanins [11]. Anthocyanin biosynthesis is a branch of flavonoid synthesis, and its biosynthetic genes are regulated by the MBW complex comprising myeloblastosis (MYB), basic helix-loop-helix (bHLH), and the WD40 repeat-containing (WD40) protein [12]. IbMYB1, IbMYB340 and ethylene-responsive transcription factor 1 (IbERF1) have been reported to regulate the anthocyanins synthesis of storage root in sweet potato [13–15].

Carotenoids are responsible for yellow, orange, and red colors found in nature, and carotenoid synthesis is initiated by geranylgeranyl diphosphate (GGPP) synthesis in the cytosolic mevalonic acid (MVA) pathway and methylerythritol 4-phosphate (MEP) pathway [16]. Plant carotenoid biosynthesis is mostly derived from the MEP route, which is only found in plastids [16]. The diverse carotenoid accumulations, such as $\alpha$-carotene, $\beta$-carotene, lutein and lycopene, are the primary contributors to a variety of carrot root colors, including orange, yellow and red [17,18]. The $\beta$-carotene, mutatochrome, phytoene, and lutein are the predominant carotenoid components found from high to low in sweet potato storage roots [19]. Carotenoid biosynthetic genes such as geranylgeranyl pyrophosphate synthase (GGPS), phytoene synthase (PSY), phytoene desaturase (PDS), $\zeta$-carotene desaturase (ZDS), carotenoid isomerase (CRTISO), lycopene $\varepsilon$-cyclase (LCY-$\varepsilon$), lycopene $\beta$-cyclase (LCY-$\beta$), $\beta$-carotene hydroxylase (CHY-$\beta$), zeaxanthin epoxidase (ZEP), and carotenoid cleavage dioxygenase (CCD) have so far been isolated and characterized in sweet potato [20]. In addition to the carotenoid pathway genes, *Orange* (Or) gene can also enhance the accumulation of carotenoids in sweet potato [21].

Sweet potato is a hexaploid heterozygote with a complex genetic background, self-pollination infertility and cross incompatibility, which make genetic linkage analysis quite difficult. GWAS has become an important and effective method for crop breeding programs. Risch first proposed GWAS for the genetic studies of complex human disease [22]. GWAS uses high-density SNPs in the mapping group to filter the molecular markers associated with complex trait performance variation based on linkage disequilibrium (LD), then analyzes its genetic effect on phenotype [23]. SNP is widely used for genetic linkage construction of molecular breeding, greatly improving the resolution and precision of the genetic map with the rapid development of next generation sequencing (NGS) techniques [24,25]. At present, GWAS has become a routine strategy for decoding genotype–phenotype associations in many species, more than 1000 studies over the last decade have used GWAS for genetic association [26].

Previously, we used specific-locus amplified fragment sequencing (SLAF-seq) technology to sequence 300 sweet potato accessions [27,28]. In this study, we analyzed the raw data of the 300 sweet potato accessions with the genome of the modern cultivar of sweet potato, 'Taizhong6', as the reference genome. Finally, we developed a large number of high-quality SNPs and integrated GWAS and transcriptomic data to identify the main genes driving SRFC differences in sweet potato.

## 2. Materials and Methods

### 2.1. Selection of Plant Material and Storage Root Phenotyping

A set of 300 sweet potato accessions, including 76 landraces and 224 modern cultivars, were evaluated in the present study. We recorded the color of the storage root flesh by eye

according to the identification standards which were shown in Table 1; SRFC was recorded (Table S1). These accessions were generated from different agro-climatic zones, and were cultivated in 2019 in the Experimental Farm of Hubei Academy of Agricultural Sciences in Ezhou District, China.

**Table 1.** Morphological traits and their criteria for sweet potato.

| Morphological Traits | Criteria for Recording |
| --- | --- |
| Storage root flesh color, SRFC | 1 = White, 2 = Light yellow, 3 = Yellow, 4 = Orange, 5 = Jacinth ikaika 6 = Light red, 7 = Red, 8 = Fuchsia, 9 = Purple, 10 = Dark purple, 11 = Black purple |

### 2.2. Read Alignment and Variation Calling

Initially, the SLAF-sequencing reads of 300 sweet potato accessions were mapped to the sweet potato reference genome (https://sweetpotao.com/ (accessed on 8 November 2021)) with BWA [29]. Then, we generated the BAM format of the mapping results and filtered the non-unique and unmapped reads with Samtools (http://samtools.sourceforge.net/samtools.shtml (accessed on 10 November 2021)) [29]. We utilized Samtools to reorder the reads, and Samtoolsflagstat was used for counting the alignment rate. The Picard package (http://broadinstitute.github.io/picard/ (accessed on 10 November 2021), version: 1.87) was applied to filter the repetitive reads. Then, the Genome Analysis Toolkit (GATK, version: 3.1-1-g07a4bf8) was applied for SNP and INDEL calling [30]. SNPs were filtered with the following parameters: QD < 2.0‖MQ < 40.0‖FS > 60.0‖SOR > 3.0‖MQRankSum < −12.5‖ReadPosRankSum < −8.0. Indel were filtered with the following parameters: QD < 2.0‖FS > 200.0 ∣ SOR > 10.0‖MQRankSum < −12.5‖ReadPosRankSum < −8.0. Finally, annotations of SNP and INDEL were performed using ANNOVAR [31]. The quality pretreatment of genotyping data was carried out for SNP call rate and MAF (minor allele frequency) with the PLINK software with thresholds (MAF < 0.05, GENO > 0.1) (http://zzz.bwh.harvard.edu/plink/tutorial.shtml (accessed on 10 November 2021)) [32].

### 2.3. GWAS Analysis

To increase the number of markers that can be tested for association with a trait, we used Beagle to perform genotype imputation [33]. GWAS was performed using the Essicient Mixed-Model Association expedited (EMMAX) software package (http://csg.sph.umich.edu//kang/emmax/download/index (accessed on 10 November 2021)) [34]. Kinship (K) matrix was also calculated using the EMMAX. The accession phenotypes were sorted using sort_pheno.pl. A mixed linear model (MLM) in EMMAX was used to test the associations. K-matrix and Q-matrix as covariates were used in the MLM to avoid spurious association [35]. The GWAS results were visually examined using Manhattan plots and quantile-quantile (QQ) plots which were drawn using R package. The *p*-value was calculated for each SNP and $-\log_{10}P > 7$ was defined as the suggestive threshold and genome-wide control threshold.

### 2.4. RNA-Seq Analysis

The transcriptomic data of root 'Xiangshu99' (white-fleshed) and 'Zhezi No1 (purple-fleshed) was found by downloading the raw sequence data from the NCBI Short Read Archive (No. PRJNA721067) [36]; the transcriptomic data of root in 'Beniharuka (BH)' (yellow-fleshed) cultivar and its 'white-fleshed mutants (WH)' was found by downloading the raw sequence data from the NCBI Short Read Archive (No. PRJDB10052) [37]. Raw data were first processed with Fastpsoftware (https://github.com/OpenGene/fastp (accessed on 20 November 2021)) to remove adaptor sequences and low-quality reads. The clean reads were mapped to the sweet potato reference genome (https://sweetpotao.com/ (accessed on 20 November 2021)) using HISAT (http://ccb.jhu.edu/software/hisat2/index.shtml (accessed on 20 November 2021)) software [38]. Htseq-count was used to quantify the expression of known genes [39]. An mRNA was considered as a DE mRNA via the

DESeq2 R package (http://bioconductor.org/packages/stats/bioc/DESeq2/ (accessed on 20 November 2021)) when it exhibited a two-fold or higher expression change and its FDR was below 0.05 in the comparisons [40]. The GO (Gene Ontology) and KEGG (Kyoto Encyclopedia of Genes and Genomes) annotation of mRNAs were conducted using EggNOG (Evolutionary Genealogy of Genes: Non-supervised Orthologous Groups), then we utilized enrichGO_pip.R and enrichKEGG_pip.R scripts to conduct GO and KEGG enrichment analyses for DE mRNAs, $p < 0.05$.

*2.5. qRT-PCR*

qRT-PCR analyses were carried out to determine the reliability of the RNA-seq results for expression profile analysis. All primers were designed according to the mRNA sequences and were synthesized commercially in the company Tianyi Huiyuan, Wuhan (Table S2). qRT-PCR for mRNAs were carried out in a 10 μL system: 1.0 μL cDNA product, 5 μL 2× qPCR mix, 3.2 μL nucleotide-free water and 0.4 μL 2.5 μM for each of the forward and reverse primers. The reactions were incubated in a BIO-RAD CFX96 for 30 s at 95 °C, followed by 40 cycles of 5 s at 95 °C, then 30 s at 60 °C. Three replications were used for all reactions and $\beta$-actin served as the endogenous reference gene. The $2^{-\Delta\Delta CT}$ method was employed to analyze the relative fold changes of genes [41].

*2.6. Statistical Analysis*

A total of 567,828 SNPs across 15 linkage groups (LG) remained after quality filtering. The 567,828 SNPs were subsequently used for further analysis. ADMIXTURE was employed to investigate population structure based on the maximum-likelihood method and generated a Q matrix [42]. A phylogenetic tree was constructed using MEGA7 based on the neighbor-joining method, visually edited using FigTree software (http://tree.bio.ed.ac.uk/software/fig- tree/(accessed on 10 November 2021)). A principal component analysis (PCA) was performed with PLINK software (www.cog-genomics.org/plink2 (accessed on 10 November 2021); v1.9). PCA plot was generated by drawing PCA.R. Linkage disequilibrium (LD) analysis was performed using PopLDdecay [43]. LD was estimated as squared allele frequency correlation ($R^2$), and only $p$-value $\leq 0.01$ for each pair of loci were considered significant. LD plot was generated by perl Plot_MultiPop.pl.

**3. Results**

*3.1. SLAF-Sequencing of 300 Sweet Potato Accessions*

In our previous study, we constructed SLAF-seq library of 300 sweet potato accessions using SLAF-seq technology and obtained a total of 498.14 Mb reads [44]. In this study, we mapped the raw data to the sweet potato reference genome with BWA, the rate ranged from 6.46% to 95.11%, with an average rate of 92.61% (Table S3). After filtering, we identified 567,828 high confidence SNPs (Table S4) (missing data < 50%, minor allele frequency (MAF) > 5%) which were used for subsequent analyses. The SNPs ranged from 29,393 (LG10) to 50,397 (LG2), and were distributed across 15 LGs of sweet potato (Figure 1), the SNP number of LG1 to LG5 was 34,856, 50,379, 30,708, 36,340, 36,129, 36,915, 43,424, 30,858, 38,001, 29,393, 47,309, 38,169, 38,107, 391,69, 38,053 in turn. We examined the distribution of polymorphisms in sweet potato genomic regions and found that 109,738, 153,623, 112,695, 99,800, and 100,673 SNPs were located within exons, introns, upstream regions (within 5 kb upstream of the transcription start sites), downstream regions (within 5 kb downstream of transcription stop sites), and intergenic regions, respectively.

*3.2. Population Structure and Genomic Variation among the 300 Sweet Potato Accessions*

The 300 sweet potato accessions included 76 landraces and 224 modern cultivars, which corresponded to roughly their geographical distributions in Asia and North America, respectively (Figure 2a). Neighbor-joining cluster analysis showed that the 300 accessions were divided into seven groups (Figure 2b); the seven groups contained 63, 23, 24, 21, 66, 56, 47 accessions, respectively. The population structure analysis and principal component

analysis (PCA) found that the 300 sweet potato accessions were clustered into 4 groups (K = 4), namely groups 1, 2, 3, and 4 (Figure 2c). Group 1, 2, 3 and 4 contain 69, 71, 83, and 77 accessions, respectively. PCA clustering showed that the accessions of the core set were distributed into two clusters: landraces and modern cultivars. Principal component 1 (PC1) represents 15.51% of the total variation, while PC2 represents 12.0% of the total variation (Figure 2d). LD analysis showed that the delay rate of landraces was faster than that of modern cultivars (Figure 2e), suggesting a high frequency of genetic recombination in landraces, further supporting the richer genetic diversity of landraces.

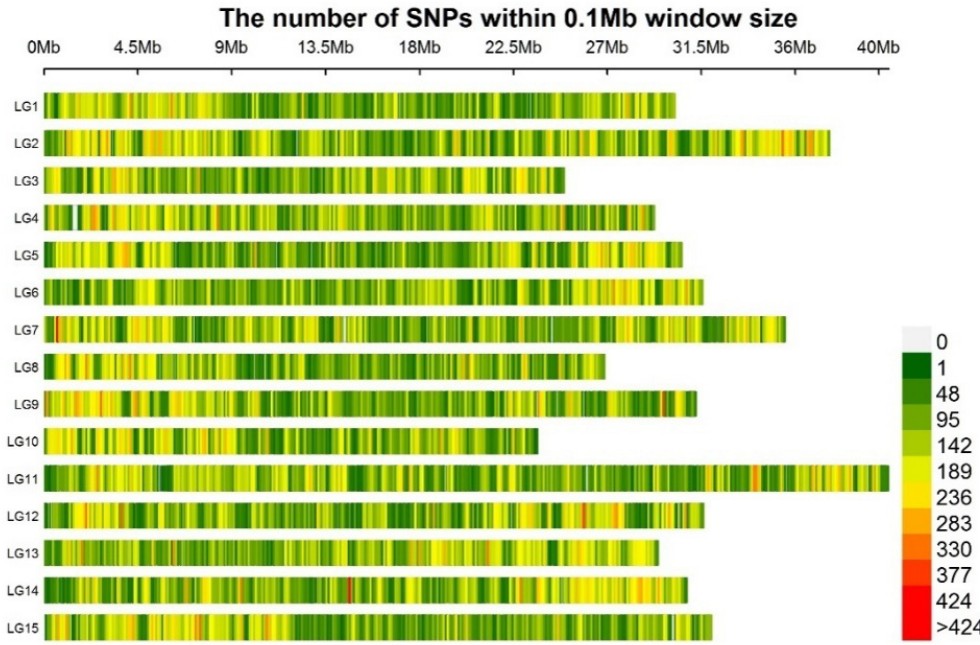

**Figure 1.** The distribution of SNPs in 15 LGs of sweet potato.

### 3.3. GWAS for Identifying Significant SNPs Associated with SRFC

SRFC is one of the most important traits associated with the quality of sweet potato. There is a correlation between the color and chemical composition of sweet potato flesh. We conducted GWAS to identify significant SNP associated with SRFC by utilizing the 567,828 SNP markers (Table S5). GWAS results were visually examined by using Manhattan plots (Figure 3a) and quantile-quantile plots (Figure 3b). The line shows the threshold for genome-wide significance ($-\log_{10}P > 7$). A total of 5 significant SNPs associated with SRFC were identified (Table 2).

Interestingly, g55964, F3′H, was identified within the 4460,608-4466,687 bp in LG14, where 1 SNP (LG14: 467,066) was strongly associated with SRFC. GWAS analysis revealed that the SNP was located at ~379 bp upstream of the start codon of the F3′H which is involved in anthocyanins biosynthesis.

g17506, BAG2-like, was identified within the 5866,570-5867,826 bp in LG5, where one SNP (LG5: 5,867,009) was strongly associated with SRFC. GWAS analysis discovered that the SNP was located in the intron region of the BAG2-like.

g25206, UGT-73D1-like, was identified within the 31,266,628-31,269,316 bp in LG6, where two SNPs (LG6: 31,267,101 and LG6: 31,267,107) were strongly associated with SRFC. GWAS analysis found that the two SNPs were located in the UTR3 of the UGT-73A-like.

g58377, VQ25 isoform X2, was identified within the 21,793,980-21,794,619 bp in LG14, where one SNP (LG14: 21,791,164) was strongly associated with SRFC. GWAS analysis showed that the SNP was located at ~2816 bp upstream of VQ motif-containing protein 25 isoform X2.

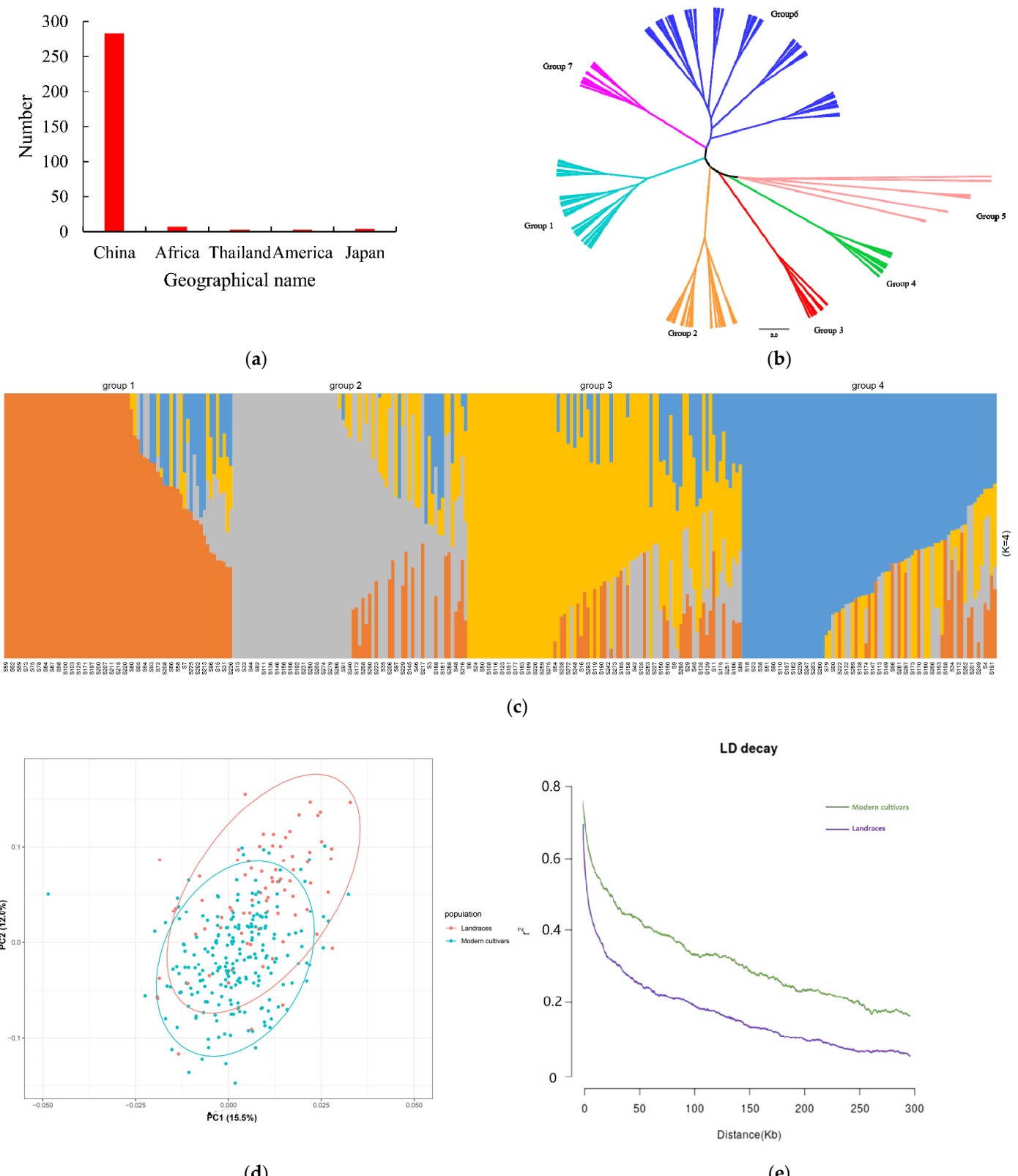

**Figure 2.** Distribution, population structure, PCA, and LD decay of the 300 sweet potato germplasm accessions. (**a**) The geographic distribution of the 300 sweet potato accessions based on country of origin. (**b**)The phylogenetic tree of 300 sweet potato accessions based on the analysis of 567,828 SNPs. (**c**) Population structure analysis using 567,828 SNPs (missing data < 50%, MAF > 5%, $r^2$ < 0.2), K = 4. (**d**) PCA plot of the first two components (PC1 and PC2) of the 300 sweet potato accessions. (**e**) Genome-wide average LD decay of the 300 sweet potato accessions.

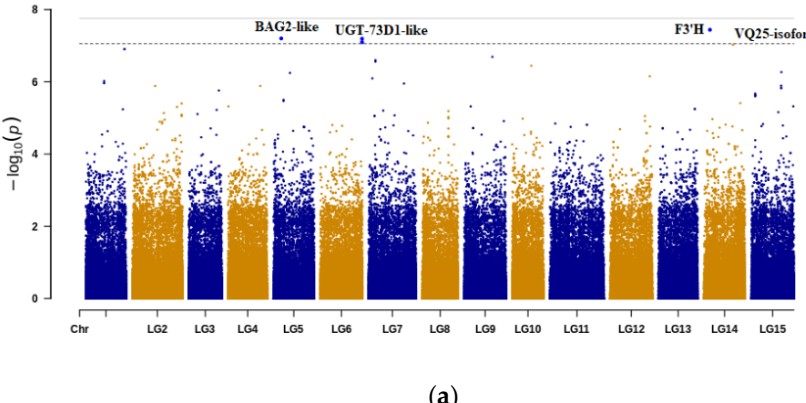

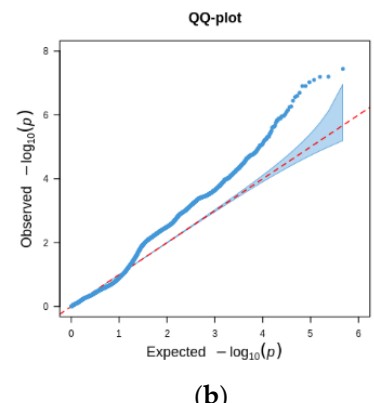

(**a**)                                           (**b**)

**Figure 3.** Genome-wide association study analysis of SRFC in 300 sweet potato accessions. (**a**) Manhattan plots of SNPs association for SRFC. (**b**) Quantile–quantile plots of expected and observed $-\log_{10}(p)$ values at an FDR cut-off $\leq 0.05$. The gray areas are the 95% confidence intervals under the null hypothesis of no association between the SNP and the trait. The red line is the expected line under the null distribution. The blue points are the observed distribution.

**Table 2.** Summary of SNPs above the significance thresholds.

| Marker Name | Candidate Genes | Annotated | *p*-Value | R2 | LG | Allele | Position |
|---|---|---|---|---|---|---|---|
| LG14: 4,467,066 | g55964 | F3′H | $3.63 \times 10^{-8}$ | 0.096366 | LG14 | A/G | 4,467,066 |
| LG5: 5,867,009 | g17506 | BAG2-like | $6.31 \times 10^{-8}$ | 0.093125 | LG5 | A/G | 5,867,009 |
| LG6: 31,267,101 | g25206 | UGT-73D1-like | $6.41 \times 10^{-8}$ | 0.093027 | LG6 | C/G | 31,267,101 |
| LG6: 31,267,107 | g25206 | UGT-73D1-like | $7.99 \times 10^{-8}$ | 0.091735 | LG6 | T/C | 31,267,107 |
| LG14: 21,791,164 | g58377 | VQ25-isoform X2 | $9.43 \times 10^{-8}$ | 0.090762 | LG14 | C/T | 21,791,164 |

### 3.4. Differentially Expressed Gene between White-Fleshed and Purple-Fleshed Sweet Potato

To study the key genes that regulate purple formation of root in sweet potato, we analyzed the transcriptomic data of root in 'Xiangshu99' (white-fleshed) and 'Zhezi No1 (purple-fleshed). In the transcriptomic analysis, the expression levels of the genes were shown in Table S6a. Compared with 'Xiangshu99', 442 upregulated and 117 downregulated DEGs in 'Zhezi No1'were detected (Table S6b). GO enrichment analysis was conducted to further understand the function of the identified DEGs. A total of 139 (Table S7a) and 64 (Table S7b) terms were significantly enriched in the upregulated and downregulated genes, with $p < 0.05$, respectively. The 20 most enriched GO terms in the upregulated and down-regulated genes are shown in Figure 4. The "flavonoid biosynthetic process" (GO: 0009813) and "flavonoid metabolic process" (GO: 0009812) were the most abundant upregulated DEGs (Figure 4a). The predominantly enriched pathways in the downregulated DEGs were the "Phenylpropanoid biosynthetic process" (GO: 0009699) and "Suberin biosynthetic process" (GO: 0010345) (Figure 4b). The KEGG enrichment analysis found that a total of five (Table S8a) and five (Table S8b) terms were significantly enriched in the upregulated and downregulated genes, with $p < 0.05$, respectively. The "Flavonoid biosynthesis" (ko00941), and "Anthocyanin biosynthesis" (ko00942) were the most abundant in upregulated DEGs (Figure 5a). The predominantly enriched pathways in the downregulated DEGs were involved in the "Protein processing in endoplasmic reticulum" (ko04141), whereas none of the metabolic pathways were significantly enriched (Figure 5b). The results of GO and KEGG enrichment implied that the flavonoid anthocyanin biosynthetic-related genes regulate the purple formation of roots in sweet potato.

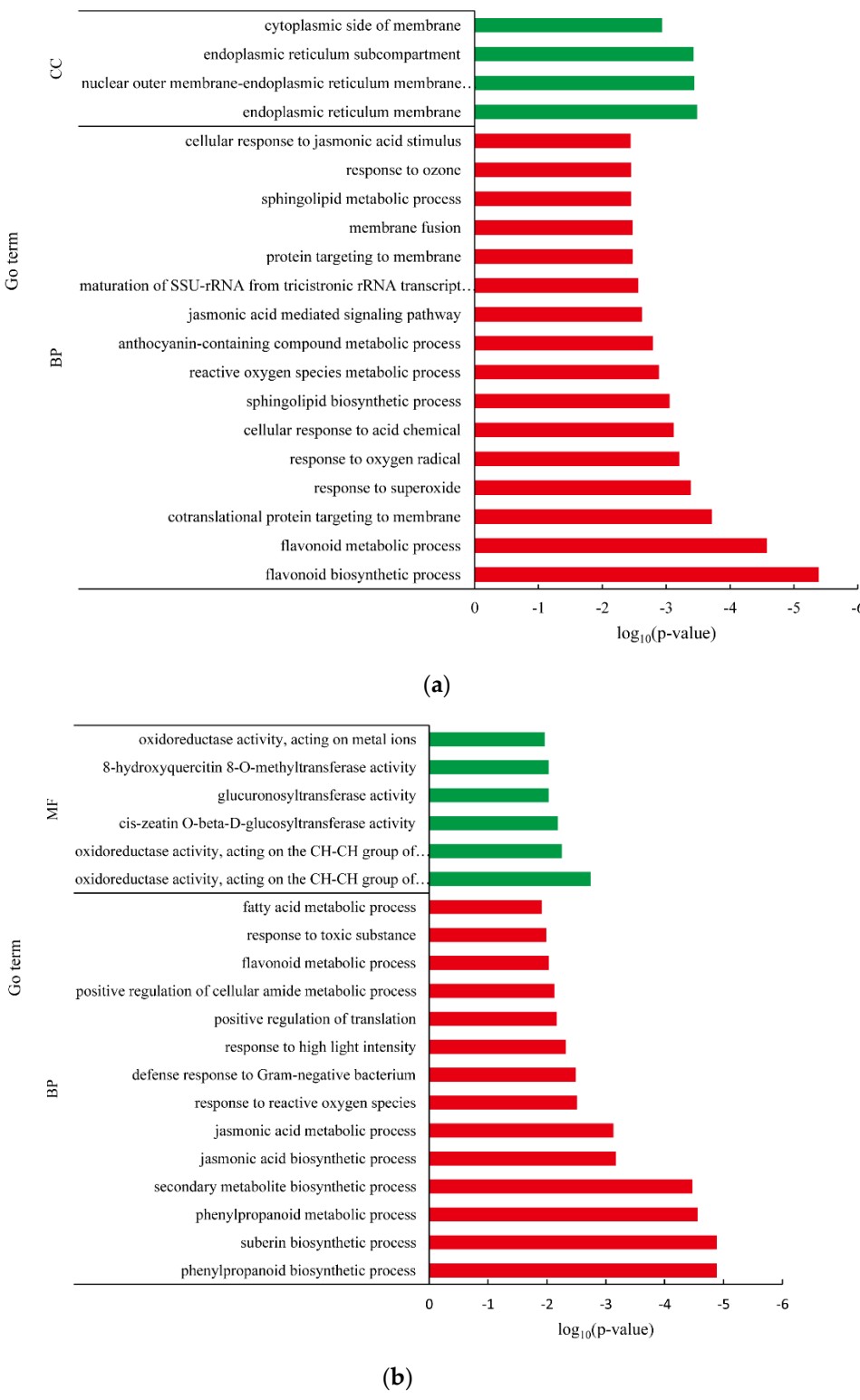

**Figure 4.** GO enrichment of DEGs in the root of Xiangshu99 vs. Zhezi No1. (**a**) upregulated genes; (**b**) downregulated genes. GO terms belonging to biological processes (BP), cellular components (CC), and molecular functions (MF).

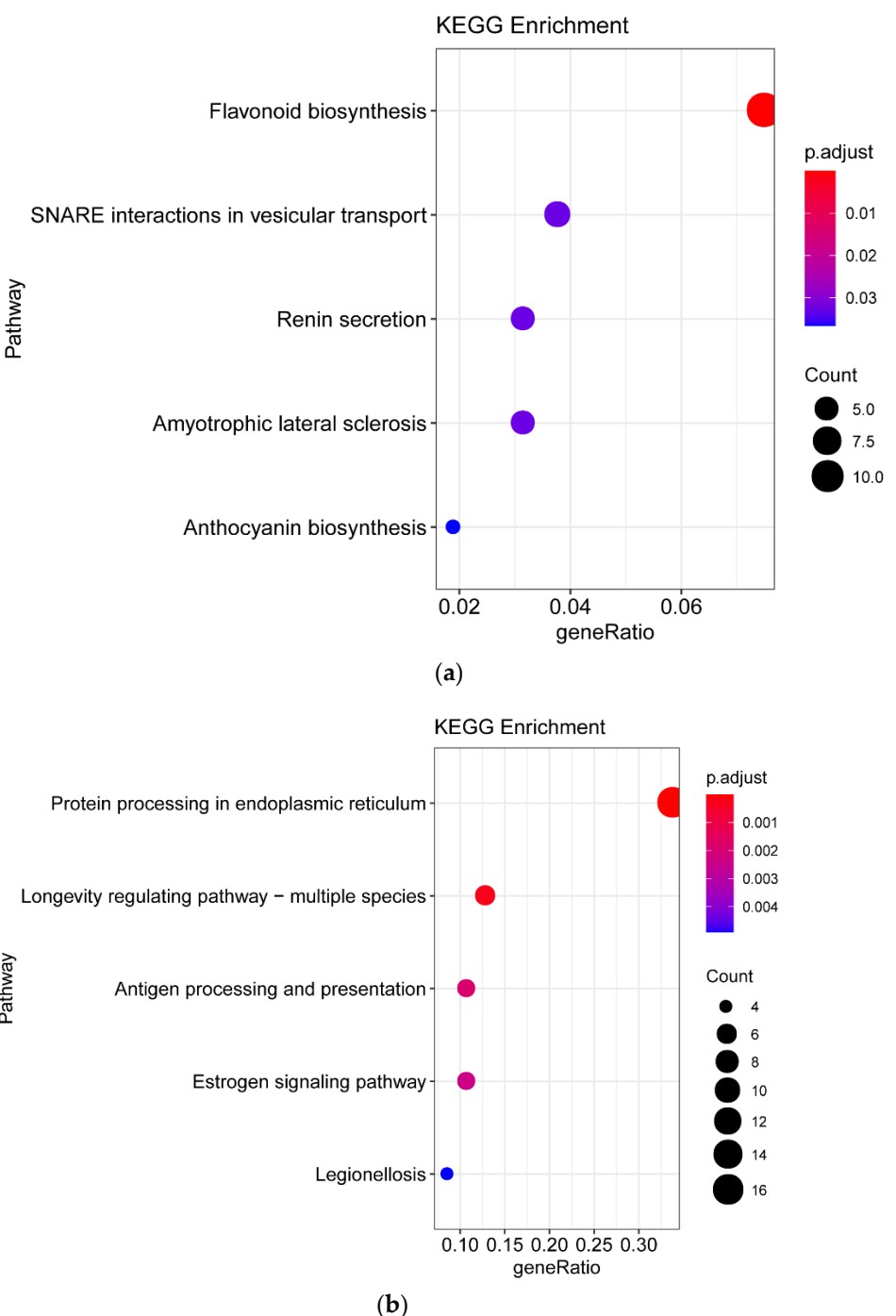

**Figure 5.** KEGG enrichment of DEGs in the root of Xiangshu99 vs. Zhezi No1. (**a**) upregulated genes; (**b**) downregulated genes.

### 3.5. Differentially Expressed Gene between White Fleshed and Yellow-Fleshed Sweet Potato

To identify the key genes involved in yellow-fleshed, we analyzed the transcriptomic data of the root in 'Beniharuka (BH)' (yellow-fleshed) cultivar and its 'white-fleshed mutants (WH)'. In the transcriptomic analysis, the expression levels of the genes were shown in Table S9a. Compared with 'WH', a total of 108 upregulated and 97 downregulated DEGs were detected in 'Beniharuka' (Table S9b). GO enrichment analysis was conducted to further understand the function of these DEGs, and a total of 75 (Table S10a) and 56 (Table S10b) terms were significantly enriched in the upregulated and downregulated genes, with a $p < 0.05$, respectively. The 20 most enriched GO terms in the upregulated

and downregulated genes were shown in Figure 6. Among the GO terms in upregulated genes, "abscisic acid transport" (GO: 0080168), "isoprenoid transport" (GO: 0046864), and "terpenoid transporter" (GO: 0046865) were significantly enriched in upregulated genes (Figure 6a). The predominantly enriched pathways in the downregulated DEGs were reported for "cadmium ion transport" (GO: 001569) (Figure 6b). The KEGG enrichment analysis found that a total of 16 (Table S11a) and one (Table S11b) terms were significantly enriched in the upregulated and downregulated genes, with $p < 0.05$, respectively. The most abundant upregulated DEGs were observed for "Glycolysis/Gluconeogenesis" (ko00010) (Figure 7a). The enriched pathways in the downregulated DEGs were involved in "Isoquinoline alkaloid biosynthesis" (ko00950) (Figure 7b).The results of GO enrichment implied that the terpenes transporter-related genes regulate the yellow formation of sweet potato root.

### 3.6. Exploring Candidate Genes for SRFC by RNA-Seq

According to the result of RNA-seq, a total of 35 GO terms were found to be associated with the two candidate genes, and four KEGG pathways were found to be associated with two candidate genes (Tables 3 and 4). GO analysis showed that g55964 belonged to "secondary metabolic process", "secondary metabolite biosynthetic process", "metabolic process", and "biosynthetic process", whereas g25206 belonged to "glucosyltransferase activity" and "UDP-glycosyltransferase activity". KEGG analysis suggested that g55964 and g25206 were involved in the "biosynthesis of secondary metabolites", signifying the involvement of g55964 and g25206 in the biosynthesis of metabolites in sweet potato.

### 3.7. Candidate Genes Expression Analysis

We quantified the expression of four candidate genes in the flesh of sweet potato varieties that varied in color by analyzing the transcriptomic data (Figure 6, Table 5). The result displayed that only g55964 displayed significant differential expression between 'Zhezi No1' and 'Xiangshu 99'. We found that the expression of g55964 in 'Zhezi No1' was 46.8-fold higher than that of 'Xiangshu99' (Table S6b), and all candidate genes had no expression in 'WH' and 'BH' (Figure 8). We performed qRT-PCR to quantify the relative expression of g55964 among eight varieties. The result of qRT-PCR showed that the expression of g55964 in purple-fleshed sweet potato was significantly ($p < 0.01$) higher than that of non-purple fleshed sweet potato (Figure 9, Table S12).

**Table 3.** GO analysis of candidate genes related to the RNA-seq results.

| Gene ID | GO |
|---------|-----|
| g55964 | GO:0016705; GO:0016020; GO:0055114; GO:0016709; GO:0004497; GO:0009733; GO:0042221; GO:0019748; GO:0016491; GO:0050896; GO:0044550; GO:0008150; GO:0008152; GO:0009725; GO:0010033; GO:0009719; GO:0003674; GO:0005575; GO:0009058 |
| g17506 | none |
| g25206 | GO:0046527; GO:0005622; GO:0080044; GO:0043226; GO:0043231; GO:0043229; GO:0044424; GO:0035251; GO:0005623; GO:0016740; GO:0016758; GO:0016757; GO:0044464; GO:0080043; GO:0003674; GO:0043227; GO:0005575; GO:0008194 |
| g58377 | none |

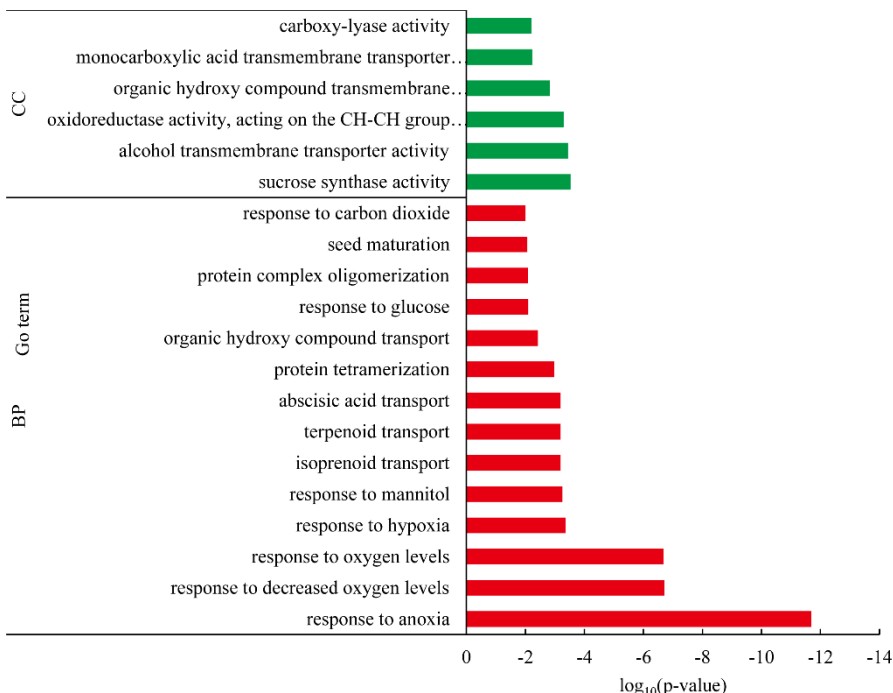

(**a**)

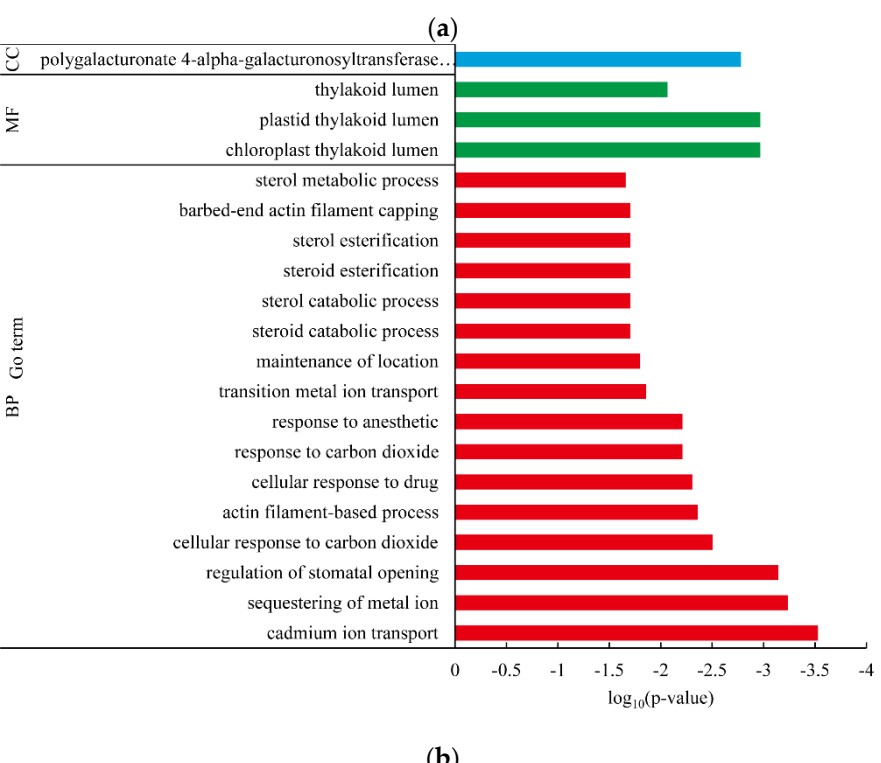

(**b**)

**Figure 6.** GO enrichment of DEGs in the root of WH3 vs. BH. (**a**) upregulated genes; (**b**) down-regulated genes. GO terms belonging to biological processes (BP), cellular components (CC), and molec-ular functions (MF).

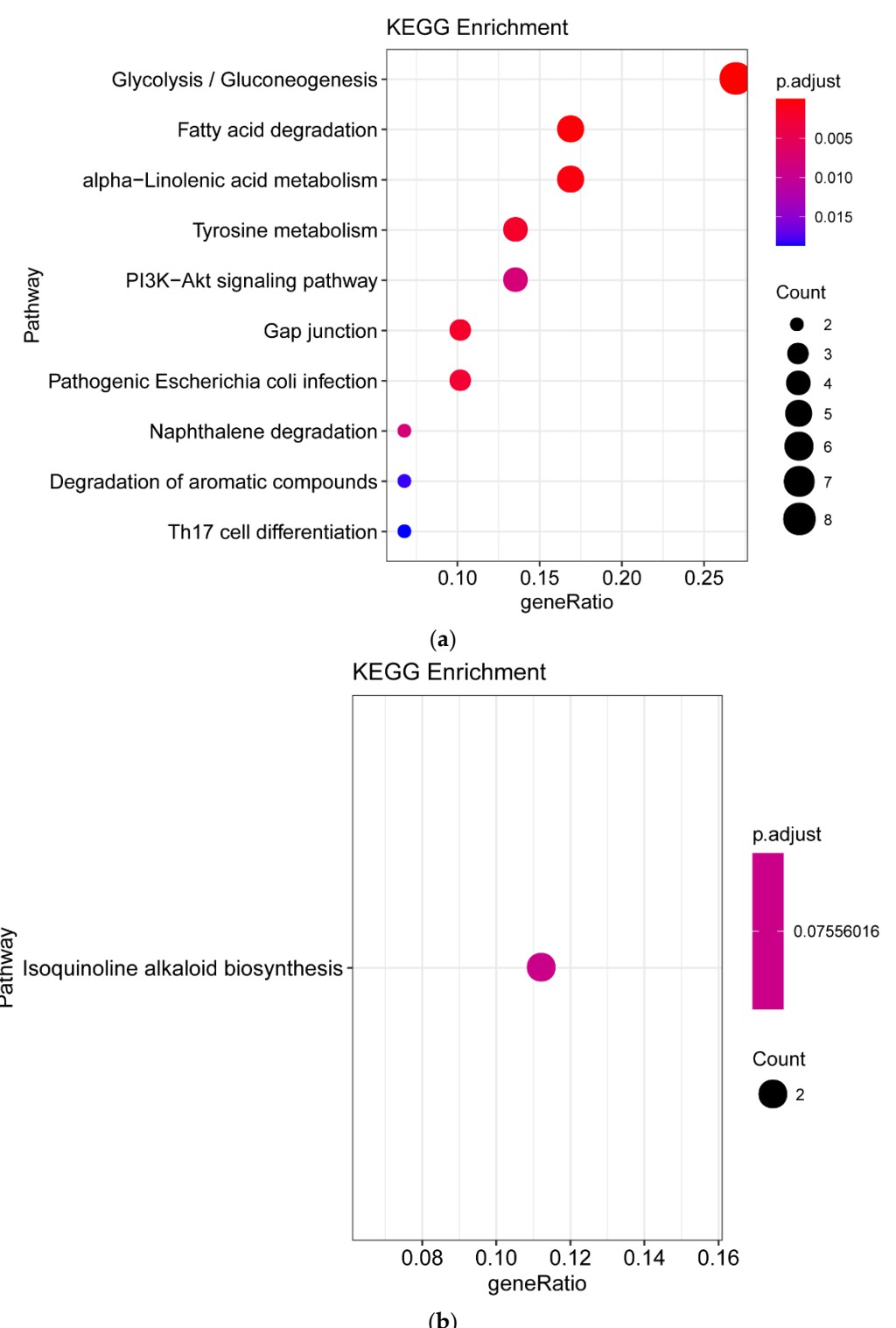

**Figure 7.** KEGG enrichment of DEGs in the root of WH3 vs. BH. (**a**) upregulated genes; (**b**) downregulated genes.

**Table 4.** KEGG analysis of candidate genes related to the RNA-seq results.

| Pathway_ID | Pathway_Name | Gene_ID | KO_Entry | EC |
|---|---|---|---|---|
| ko00941 | Flavonoid biosynthesis | g55964 | K05280 | EC:1.14.14.82 |
| ko00944 | Flavone and flavonol biosynthesis | g55964 | K05280 | EC:1.14.14.82 |
| ko01100 | Metabolic pathways | g55964 | K05280 | EC:1.14.14.82 |
| ko01110 | Biosynthesis of secondary metabolites | g55967 | K05280 | EC:1.14.14.82 |
| ko01110 | Biosynthesis of secondary metabolites | g25206 | K13496 | EC:2.4.1. |

**Table 5.** The FPKM value of genes in varieties.

| | Xiangshu99 | Zhezi No1 | BH | WH |
|---|---|---|---|---|
| g55964 | 0.46 | 16.86 | 0.00 | 0.00 |
| g17506 | 11.17 | 15.35 | 0.00 | 0.00 |
| g25206 | 0.11 | 0.00 | 0.00 | 0.00 |
| g58377 | 0.12 | 0.00 | 0.00 | 0.00 |

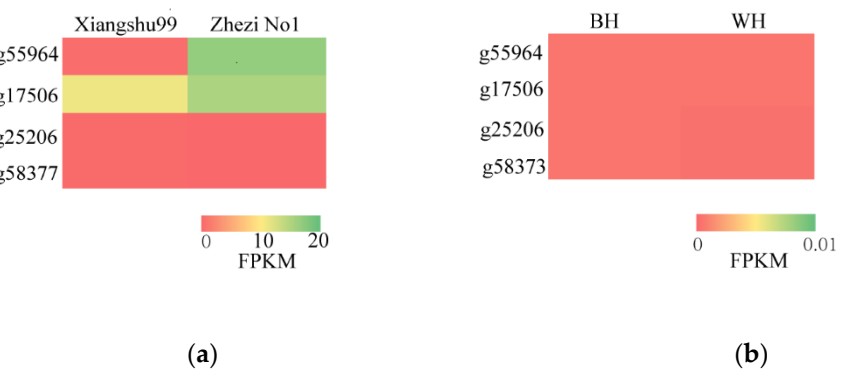

(**a**)            (**b**)

**Figure 8.** Heatmap of candidate genes in storage root by RNA-seq. (**a**) The expression of candidate genes in 'Xiangshu99′ and 'Zhezi No1′ (No. PRJNA721067); (**b**) The expression of candidate genes in 'BH' and 'WH' (No. PRJDB10052).

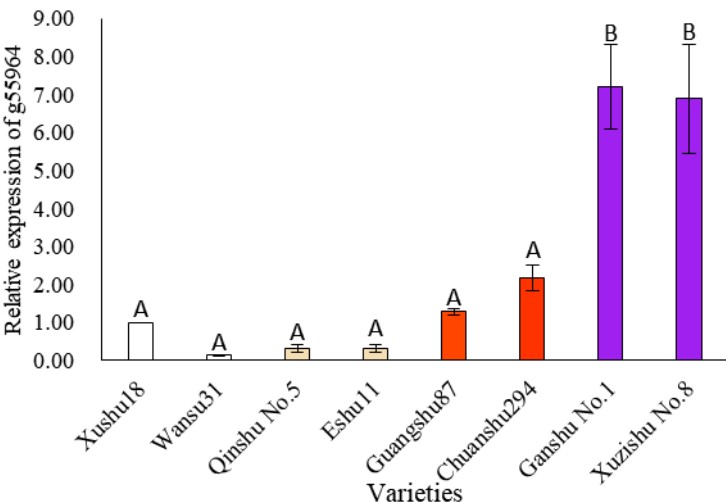

**Figure 9.** g55964 expression comparisons between different colors of flesh by RT-qPCR. The SRFC in 'Xushu18′and 'Wansu31′are white, the SRFC in 'Qinshu No.5′and 'Eshu11′are light-yellow, the SRFC in 'Xiangshu99′, 'Guangshu87′, and 'Chuanshu294′ are jacinth, the SRFC in 'Zhezishu No.1′, 'Ganshu No.1′ and 'Xuzishu No.8′ are purple.

## 4. Discussion

### 4.1. Genetic Variation among the 300 Accessions

In our previous study, we sequenced and characterized the genetic diversities of 300 natural accessions of sweet potato by SLAF-seq, most of which were from major sweet potato production regions (i.e., America, Africa, China, Japan, and Korea) [28]. In this study, population-genetics analyses of the SNPs were generated by mapping reads to hexaploid sweet potato reference genomes, the 'Taizhong6'. Based on population structure and PCA, 300 sweet potato accessions were clustered into four groups. The faster LD decay rate of landraces was faster than that of modern cultivars, suggesting that the genetic diversity of landraces was higher than modern cultivars. The result was consistent with He et al. [45], who reported that sweet potato landraces possess high genetic diversity and that the genetic diversity of modern cultivars is low.

### 4.2. GWAS Analysis of SRFC in Sweet Potato

In order to mine the candidate genes for SRFC, we recorded the SRFC of 300 sweet potato accessions, and we used the SNPs site to conduct GWAS of SRFC. We identified five potential SNPs which were associated with SRFC ($-\log_{10}P > 7$), and linked to four candidate genes. g55964 was annotated as F3'H, catalyzing naringenin and dihydokaempferol, and generated eriodictyol and dihydroquercetin which are important intermediates in the biosynthesis of anthocyanins and procyanidins [46]. We found that g25206 was annotated as UGT-73D1-like. It has been reported that uridine diphosphate glycosyltransferase (UGT) regulates the bioactivity, solubility, and transportation of receptors within the cells through modulating glycosyl transfer reaction [47]. UGT can catalyze flavonols and anthocyanins into corresponding glycosides [48,49]. g17506 was annotated as BAG2-like. Previously, it was reported that BCL2-associated athanogene (BAG) family members can regulate plant stress and development [50]. To the best of our knowledge, the BAG family has not been reported to be involved in metabolite synthesis. The g58377 was annotated as VQ25-isoform X2. The VQ motif-containing proteins (VQ) can interact with the WRKY transcription factor through the conserved VQ domain and plays an important role in the signal pathway mediated by the WRKY transcription factor [51,52]. The WRKY transcription factor can control the secondary metabolites synthesis, such as phenols, terpenes, and alkaloids [53]. Among them, anthocyanins and carotenoids are the two main pigments in sweet potato; it is not clear whether VQ can interact with WRKY transcription factors to regulate the synthesis of anthocyanins and carotenoids. Whether BAG and VQ are involved in metabolite synthesis remained to be studied.

### 4.3. Functional Analysis of DEGs in Different SRFC Varieties

The GO and KEGG enrichment result showed that flavonoid and anthocyanin biosynthetic-related genes regulated the purple formation of the root in sweet potato. Li et al. also reported that flavonoid and anthocyanin biosynthesis were related to the flesh color of storage root [54]. The g55694 was annotated into the flavonoid biosynthesis pathway, and its expression in purple-fleshed sweet potato was much higher than that of non-purple fleshed sweet potato. Thus we speculated that g55694 was the one of the key genes involved in the purple formation of the root in sweet potato.

### 4.4. Quantitative Trait Loci Analysis of Flesh Color in Sweet Potato

Previously, some studies have reported that QTLs significantly associated with flesh color in sweet potatoes are located on LG3 and LG12. Zhang et al. [55] performed GWAS for flesh color in purple sweet potato and identified one QTL for *IbMYB1-2* on LG12. Gemenet et al. [56] identified two major QTLs that affect β-carotene and flesh color; they are *phytoene synthase* and Orange gene located within QTLs on LG3 and LG12, respectively. Yan et al. [57] identified ten QTLs for flesh color, skin color, and anthocyanin content on LG12 based on the linkages map. In this study, we identified one QTL for flesh color on LG14, the QTL was linked to F3'H which was involved in flavonoid biosynthesis.

Anthocyanins and carotenes are responsible for sweet potato storage root colors, with complex components, which are affected by various environmental factors. Thus, further research is required to dissect the genomic mechanisms controlling different types of anthocyanins and carotenes.

## 5. Conclusions

We recorded SRFC of 300 sweet potato accessions and conducted GWAS in sweet potato for SRFC. The study identified five unique SNPs which were significantly associated with SRFC trait, and linked to four genes. The transcriptome analyses showed that anthocyanin is responsible for the formation of purple roots in sweet potato, while carotenoid is responsible for the formation of yellow roots. We also employed GWAS and transcriptomic analyses to identify one QTL for flesh color on LG14, and the QTL was linked to g55964 which was involved in flavonoid biosynthesis.

**Supplementary Materials:** The following supporting information can be downloaded at: https://www.mdpi.com/article/10.3390/agronomy12050991/s1. Table S1: The information of 300 accessions; Table S2: The primer sequencing information of genes for qRT-PCR; TableS3: The mapped rate of 300 accessions' raw data to the sweet potato reference genome; Table S4: The information of 567,828 SNPs; Table S5: The result of GWAS for identifying significant SNP associated with SRFC; Table S6a: The expression level of genes in 'Xiangshu99' and 'Zhezi No1'; Table S6b: The list of DEGs between 'Xiangshu99' and 'Zhezi No1'; Table S7a: The GO enrichment analysis of up-regulated DEGs between 'Xiangshu99' and 'Zhezi No1'; Table S7b: The GO enrichment analysis of down-regulated DEGs between 'Xiangshu99' and 'Zhezi No1'; Table S8a: The KEGG enrichment analysis of up-regulated DEGs between 'Xiangshu99' and 'Zhezi No1'; Table S8b: The KEGG enrichment analysis of down-regulated DEGs between 'Xiangshu99' and 'Zhezi No1'; Table S9a: The expression level of genes in 'WH' and 'BH'; Table S9b: The list of DEGs between 'WH' and 'BH'; Table S10a: The GO enrichment analysis of up-regulated DEGs between 'WH' and 'BH'; Table S10b: The GO enrichment analysis of down-regulated DEGs between 'WH' and 'BH'; Table S11a: The KEGG enrichment analysis of up-regulated DEGs between 'WH' and 'BH'; Table S11b: The KEGG enrichment analysis of down-regulated DEGs between 'WH' and 'BH'; Table S12: The relative expression of g55964 in eight varieties.

**Author Contributions:** C.J. and X.Y. designed the study. Y.L. and R.P. analyzed the data and prepared the manuscript. W.Z., L.W., J.L., S.C. and X.J. provided manuscript revision services. All authors have read and agreed to the published version of the manuscript.

**Funding:** This work was supported by the National Key R&D Program of China (2018YFD1000700, 2018YFD1000705); China Agriculture Research System (CARS-11-C-15); the Characteristic Discipline of Hubei Academy of Agricultural Sciences (2015TSXK06).

**Institutional Review Board Statement:** Not applicable.

**Informed Consent Statement:** Not applicable.

**Data Availability Statement:** Not applicable.

**Conflicts of Interest:** The authors declare no conflict of interest.

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
