# Peer review of "Integrating Genome-Wide Association Study with Transcriptomic Analysis to Predict Candidate Genes Controlling Storage Root Flesh Color in Sweet Potato"

_agronomy, doi:10.3390/agronomy12050991_

Round 1
Reviewer 1 Report
This manuscript uses GWAS to predict candidate genes for flesh color in sweet potato. It seems like they used the previously published data for this study but performed new analysis and qRT-PCR for candidate genes. This manuscript should be intensively revised.
Major comments
In Introduction, the authors should include information about sweet potato’s self-pollination infertility, and cross incompatibility mentioned in Abstract.
In Methods, the authors need to specify how they recorded the color trait, by eye or instrument. They used the previously published data which are not cited at all in Methods. Also, the authors need to present scripts used in data analysis, for example, sort_pheno.pl and Plot_Multipop.pl.
If the authors perform qRT-PCR for four candidate genes, they should describe the results.
In Results, lots of part should be moved to Methods section.
In Discussions, the authors describe the results that should be included in Results. There is no biological inference for their result.
Minor comments
Abstract: none-purple è non-purple
In text, do not use “chromosome” since they are Linkage Group (LG), not chromosome.
Lines 58 – 60: need to present the full name of genes
Line 86: What is SLAF-seq?
Lines 97 – 99: the same information is presented twice. Remove redundancy.
Lines 113 – 114: Indel was repeated twice. Is it correct?
Line 121: To increases è To increase
Line 128 Man- hattan è Manhattan
Figure 2 needs to be rearranged. (a) and (b) should be horizontally paralleled, not vertically. Also, need to describe four groups in Figure 2b and need to cite Figure 2e in text.
Table 2. Adjust column width to have words in one line. Since all five genes are related to SRFC, the first column can be removed, and add that information in a legend. Market è Marker
Lines 218 – 220 & 223 - 226: this should be included in Materials and Methods
Line 226: FDR should be used instead of p-value.
Figure 4 & 6. The words are not readable. The authors should use a higher resolution of image. Legend is not correct.
Figure 5 & 7. Same comment as Figure 4 & 6. Also, how to draw a dot plot should be described in M&M.
Figure 8. There are only two contrasting samples for four genes, so it would be better to present a table with FPKM values.
Figure 9. Suggestion – change the color of bar according to the flesh color.
Reviewer 2 Report
Suggestions for author:
- Table 2. Summary of SNPs above the significance thresholds: R squared values are missing, which are considered as the most important parameter of Marker trait associations, it indicates how much an SNP contributes to the trait it is associated with.
- Authors should provide a chromosome wise summary of SNPs after filtering the raw sequencing data. Along with the Figure 1. (The distribution of SNPs in 15 chromosomes of sweet potato).
- Sometimes authors have given space between a value and unit sometimes don’t. Kindly follow a uniform pattern
- Line no. 71, 72, 73, 74, 80, 119, 123, 124: there must be a space before brackets.
- Line no. 82: it is Next Generation Sequencing, not Next General Sequencing. Line no. 128: it is Manhattan, not Man-hattan
Round 2
Reviewer 1 Report
Thank you for responding to my comments.
I still have two questions for this paper.
1) Why didn’t you perform qRT-PCR for g17506 that shows high expression in two accessions?
2) "To the best of our knowledge, the BAG family has not been reported to be involved in metabolite synthesis" => It sounds like you report that the BAG family is involved in metabolite synthesis, but your data does not support this inference.
3) There are many grammatical errors in this manuscript.
